

# Quantifying the mass loading of particles in an ash cloud remobilised from tephra deposits on Iceland

Frances Beckett[1], Arve Kylling[2], Guðmunda Sigurðardóttir[3], Sibylle von Löwis[3], and Claire Witham[1]

[1]Met Office, Exeter, UK
[2]NILU-Norwegian Institute for Air Research, Oslo, Norway
[3]Icelandic Meteorological Office, Reykjavik, Iceland

*Correspondence to:* Frances Beckett (frances.beckett@metoffice.gov.uk)

**Abstract.**

On the 16–17 September 2013 strong surface winds over tephra deposits in southern Iceland led to the resuspension and subsequent advection of significant quantities of volcanic ash. The resulting resuspended ash cloud was transported to the southeast over the North Atlantic Ocean and, due to clear skies at the time, was exceptionally well observed in satellite imagery. We

use satellite based measurements in combination with radiative transfer and dispersion modelling to quantify the total mass of ash resuspended during this event. Typically ash clouds from explosive eruptions are identified in satellite measurements from a negative Brightness Temperature Difference (BTD) signal, however this technique assumes that the ash resides at high levels in the atmosphere. Due to a temperature inversion in the troposphere over southern Iceland during the 16 September 2013 the resuspended ash cloud was constrained to altitudes of $< 2$ km asl. We show that a positive BTD signal can instead

be used to identify ash-containing pixels from satellite measurements. The timing and location of the ash cloud identified using this technique from measurements made by VIIRS on-board the Suomi satellite agree well with model predictions using the dispersion model NAME. Total column mass loadings are determined from the VIIRS data using an optimal estimation technique which accounts for the low altitude of the resuspended ash cloud and are used to calibrate the source strength in the resuspended ash scheme in NAME. Considering the tephra deposits from the recent eruptions of Eyjafjallajökull and Grímsvötn

we estimate that $\sim 0.2$ Tg of ash was remobilised during this event.

## 1 Introduction

Iceland is one of the most active volcanic regions on Earth, with $\geq 20$ eruptions per century (Thordarson and Höskuldsson, 2008), and explosive eruptions can leave behind widespread ash deposits (e.g. Larsen et al., 2001; Carey et al., 2010; Jude-Eton et al., 2012). These deposits are subject to intense aeolian processes; Iceland is windy and the lack of vegetation inhibits soil

formation and particle binding, resulting in significant remobilsation events in the years following a volcanic eruption (Arnalds et al., 2016). The eruptions of Eyjafjallajökull in 2010 and Grímsvötn in 2011 provided a fresh source of unconsolidated ash deposits in southern Iceland and there have been a number of significant resuspended ash events in the years following these eruptions (Thorsteinsson et al., 2012; Arnalds et al., 2013). Following a blizzard on the 6 March 2013 resuspended ash was deposited in Reykjavik, and particles were identified as having originated from both the Eyjafjallajökull 2010 and Grímsvötn



2011 deposits (Liu et al., 2014). Resuspended 'ash storms' can pose a significant hazard to the local population; decreased visibility levels impact ground transportation and airports (Guffanti et al., 2009; Liu et al., 2014) and poor air quality episodes can be a concern for human health (e.g. Horwell and Baxter, 2006) and livestock (Wilson et al., 2011).

Following the eruption of Eyjafjallajökull in 2010, which deposited $140 \pm 20 \times 10^6$ m$^3$ of tephra in Iceland (Gudmundsson et al., 2012), the Met Office in the UK has provided routine forecasts to the Icelandic Meteorological Office (IMO) which indicate the likely timing and location of resuspended ash clouds. Forecasts are produced using the Lagrangian atmospheric dispersion model NAME (Numerical Atmospheric-dispersion Modelling Environment, (Jones et al., 2007)), which includes a resuspension scheme developed by Leadbetter et al. (2012). Resuspended particles are advected by 3-dimensional winds pro-
vided by the Met Office's Numerical Weather Prediction (NWP) model, and dispersed using random walk techniques which account for turbulent structures in the atmosphere.

The emission of remobilised particles depends on the meteorological conditions, soil moisture, terrain roughness and the characteristics of the fallout deposit, including the size and density of particles and deposit thickness (Gillette and Passi, 1988).
NAME includes a dust scheme which explicitly models the resuspension of mineral particles; the emission rate and the size distribution of the resuspended particles is calculated as a function of soil moisture, vegetation fraction, clay fraction and the wind friction velocity (Woodward, 2001; Athanassiadou et al., 2006). However, information on the spatially varying surface characteristics of ash deposits is often not available, especially when the deposits are relatively recent (Leadbetter et al., 2012; Folch et al., 2014). Instead Leadbetter et al. (2012) implemented a simple emission scheme in NAME for resuspended vol-
canic ash in which remobilsation occurs when the local wind friction velocity exceeds a prescribed threshold and precipitation rates are low. Emission rates were calibrated using measured PM$_{10}$ data collected at multiple sites across Iceland from two significant resuspension events on the 23 May – 2 July 2010 and the 21 September 2010 – 16 February 2011, shortly after the eruption of Eyjafjallajökull in 2010. However, in the following year the eruption of Grímsvötn resulted in further widespread tephra deposits (Hreinsdöttir et al., 2014), providing an additional source of remobilised ash which is not accounted for in the
calibration presented in Leadbetter et al. (2012). It is also expected that the scaling coefficient used to calculate emission rates of resuspended ash in the Leadbetter et al. (2012) approach will vary with time as deposits are dispersed, eroded and compacted.

On 16–17 September 2013 strong surface winds over tephra deposits in southern Iceland led to the resuspension and subsequent advection of significant quantities of volcanic ash particles. The resuspended ash cloud was transported to the south-east
over the North Atlantic Ocean and, due to clear skies at the time, was exceptionally well observed in satellite imagery. Here we use satellite based measurements in combination with radiative transfer modelling to quantify the total column mass loadings of the resuspended ash cloud. These are then used to calibrate the emission rate applied in the resuspension scheme in NAME, from which we calculate the total mass of ash resuspended during this event.



Dust and volcanic ash may be detected by satellite instruments sensitive to either solar or thermal radiation. Infrared (IR) detection of ash clouds and retrieval of ash cloud properties have been described by, for example, Prata (1989); Wen and Rose (1994); Francis et al. (2012) and Prata and Prata (2012). Gu et al. (2003) used IR bands 31 and 32 of the Moderate Resolution Imaging Spectroradiometer (MODIS) to detect and quantify a sandstorm in China and the solar channels of MODIS are rou-

tinely used to produce aerosol charts (Remer et al., 2005). We analyse data from the Visible Infrared Imaging Radiometer Suite (VIIRS) on board the Suomi National Polar-orbiting Partnership (Suomi NPP) satellite. The brightness temperature difference between VIIRS bands M15 and M16, $BTD_V = BT_{M15} - BT_{M16}$, can be used to detect volcanic ash using an approach similar to that applied to MODIS bands 31 and 32 (Watson et al., 2004; Novak et al., 2008; Corradini et al., 2008). The $BTD_V$ signal depends on a number of factors including the properties of the ash particles (their size and shape), the altitude of the ash

cloud, and the temperature of the Earths surface (Prata and Grant, 2001). Dispersed ash following the eruption of a volcano often resides at high altitudes in the atmosphere giving a negative $BTD_V$ signal, compared to ice clouds which give positive $BTD_V$ values. In this study we explore how to identify low altitude resuspended ash clouds using the split window method.

The manuscript is organised as follows. In Section 2 observations from the event are presented: meteorological, particulate air

concentrations from an Optical Particle Counter (OPC) and satellite imagery. In Section 3 the radiative transfer and dispersion modelling is described. In Section 4 we attempt to quantify the total mass of ash resuspended during 16–17 September 2013 by calibrating the resuspension scheme in NAME with the satellite retrieved total column mass loadings. We discuss the results in Section 5 before the conclusions are presented in Section 6.

## 2 Observations

### 2.1 Meteorology

During the 16–17 September 2013 strong winds prevailed over southern Iceland. Surface wind speeds of up to 25 m s$^{-1}$ were recorded at weather stations located at Skarðsfjöruviti and Mýrdalssandur, close to the ash deposits from the eruptions of Eyjafjallajökull 2010 and Grímsvötn 2011 (Fig. 1). Wind direction data retrieved from radiosonde ascents at 12:00 UTC at Keflavík airport indicate that on the 16 September near-surface winds were north-westerly, veering north-easterly by the 17

September (Fig. 2). Temperature profiles from the ascents show that there was a temperature inversion at 850 hPa (∼1500 m asl) on the 16 September. This is also observed in the profile from the 17 September, although it is now weaker.

### 2.2 Optical Particle Counters

Increased PM$_{10}$ concentrations were recorded by an OPC located at Maríubakki during the 16–17 September 2013. The flow rate sampled by the OPC is 1 L min$^{-1}$ and particle concentrations are calculated from the count data by assuming that particles

are spherical and have a density of 2300 kg m$^{-3}$. Figure 3 shows the time series of calculated particle concentrations. The peak concentration of $1.44 \times 10^{-4}$ g m$^{-3}$ occurs at 09:00 UTC on the 17 September 2013. The observed air concentrations are



lower than those recorded by $PM_{10}$ monitors during resuspension episodes in 2010 following the eruption of Eyjafjallajökull, which typically ranged between $10^{-4} - 10^{-3}$ g m$^{-3}$ (Leadbetter et al., 2012). The lower mass loadings recorded during this event perhaps reflect the location of the OPC, which was not positioned under the main axis of the resuspended ash cloud, but instead was located at the edge of the plume as indicated in Fig. 1a.

## 2.3 Satellite Imagery

The Suomi NPP satellite including VIIRS (http://npp.gsfc.nasa.gov/viirs.html) was launched on the 28 October 2011, and placed in a sun-synchronous orbit at an altitude of ∼842 km. VIIRS has 22 bands in the solar and thermal parts of the spectrum and the bands used in this study are listed in Table 1. The spatial resolution of VIIRS is band dependent: the M3-M5 bands

have a spatial resolution of 0.742 km×0.259 km (downtrack × crosstrack) at nadir (1.60 km×1.58 km at end of scan) whilst the M15 and M16 bands have a spatial resolution of 0.742 km×0.776 km at nadir (1.60 km×1.58 km at end of scan). The M15 and M16 infrared bands have prelaunch measured noise equivalent delta temperatures ($NE\Delta T$) of 0.028 and 0.036 K respectively.

From the visible channels (M3, M4 and M5) "true" colour images can be produced during the day-time. In order to have data during the night-time as well the infrared bands (M15 and M16) are used. The brightness temperature in the M15 and M16 bands varies with the amount of water vapour in the atmosphere, the atmospheric temperature profile and the temperature of the underlying surface. For these parameters analysis data from the European Centre for Medium-Range Weather Forecasts (ECMWF) were utilized, see Supplementary Figs. S1–S4.

There are approximately 6 VIIRS overpasses over Iceland every day, typically 3 during the daytime and 3 at night. A list of all the night and daytime overpasses used in this study is given in Table 2. Note that the overpass at 01:47 UTC on the 17 September 2013 is not included in the analysis as it contained no clear ash signal and the study area was on the edge of the swath. Figure 4 shows the RGB composites from the daytime overpasses during the 16–17 September 2013, the resuspended

ash cloud is clearly observed and shown to be dispersing over the North Atlantic to the south-east. The M15 brightness temperatures for the day and night-time overpasses are shown in Supplementary Fig. S5.

## 3 Modelling

### 3.1 Dispersion Model Forecasts

The atmospheric dispersion model NAME includes a scheme to model the resuspension of volcanic ash (Leadbetter et al., 2012). Particles are remobilized from the surface when the local friction velocity ($U*$), which characterizes the wind shear





at the surface, exceeds a threshold friction velocity ($U*_t$). The threshold friction velocity depends on the properties of the particles (their size and density) and on the surface conditions; such as soil moisture and roughness, and vegetation cover. Information on the spatially varying characteristics of volcanic ash deposits is often unavailable, particularly as deposits change with time due to erosion, compaction and remobilisation. Here we follow Leadbetter et al. (2012) and Folch et al. (2014) and

take the threshold friction velocity to be 0.4 m s$^{-1}$, and assume that resuspension does not occur when precipitation rates are > 0.01 mm hr$^{-1}$. The meteorological fields used in this study are provided by the NAE (North Atlantic and European) configuration of Met Office's Unified Model (UM) (Davies et al., 2005), which has a horizontal resolution of 12 km (Bush et al., 2006).

We consider the deposits from the eruptions of Eyjafjallajökull 2010 and Grímsvötn 2011 to be potential sources of resus-

pended ash. The extent of the Eyjafjallajökull 2010 ash is based on a deposit map provided by Gudmundsson et al. (2012). In the absence of a published map of Grímsvötn deposits we use a modelled deposit, generated using NAME to simulate the eruption of Grímsvötn 2011, as described by Liu et al. (2014). All regions where ash has a depth > 5 mm are considered and the source areas used are indicated in Fig. 5. Source regions are represented in NAME by a horizontal grid with a resolution of 0.01° longitude and 0.01° latitude. The driving meteorology is considered at each grid cell in order to determine whether

particles should be resuspended.

Where resuspension occurs model particles are released with a uniform distribution between 0–10 m above the ground and are assigned a density of 2300 kg m$^{-3}$. Their size distribution depends on the source: the particle size distribution (PSD) of the Eyjafjallajökull 2010 ash is based on measurements of samples collected from deposits on 15 April 2010 (Gislason et al., 1993), 

whilst the PSD of the Grímsvötn ash is based on samples collected from deposits on the 22 May 2011 (Olsson et al., 2013). To be able to compare the modelled ash cloud to the OPC measurements and the satellite retrievals we only model particles with diameters between 1–10 $\mu$m (Kylling et al., 2014; Stevenson et al., 2015). The rate at which particles are remobilised is proportional to the cube of the excess friction velocity:

$$F = K(U* - U*_t)^3 \tag{1}$$

where $K$ is a dimensional constant used as a scaling coefficient. Without calibration $K$ is set to 1, the source strength increases as $U*$ increases and modelled air concentrations indicate areas of high and low concentrations, but the results are not quantitative. Once released into the model atmosphere particles are advected using the 3-dimensional NAE model winds and dispersed using random-walk techniques which account for turbulent structures in the atmosphere. Particles are removed from the atmosphere by both dry and wet deposition processes (Webster and Thomson, 2011, 2014). A discussion on the uncertain-

ties associated with the model set-up, the source areas, precipitation thresholds, and source mixing by previous remobilization can be found in Liu et al. (2014).



We ran NAME for the period 9 September to 2 October 2013. Figure 3 compares the time series of calculated air concentrations from OPC count data to the un-calibrated modelled particle concentrations at Maríubakki. The modelled peak concentration is at 19:00 UTC on 16 September 2013, ∼19 hours earlier than the recorded peak concentration by the OPC at 09:00 UTC on 17 September 2013. A possible explanation for this time-lag between the modelled and observed peaks could

be because resuspension is suppressed in NAME when precipitation rates are $> 0.01$ mm hr$^{-1}$. This approach does not account for the time required to wet the deposit and prevent resuspension, and to dry the deposit before resuspension can restart. However, comparing the particle concentrations from the OPC count data and the un-calibrated model output to the NAE precipitation rates and local friction velocity at Maríubakki we show that there was no precipitation in Maríubakki during the 15–17 September indicating that during the 24 hours prior to the modelled peak concentration the deposit was dry (Figs. 6a

and 6b). Therefore it is unlikely that the offset in the modelled and observed peak concentrations can be ascribed to the lack of parameterization for a drying-out process in NAME. Figures 6c and 6d show that the peak in the OPC data does not correspond well with the peak in the modelled friction velocity ($U*$). This suggests that a significant fraction of the resuspended ash particles detected by the OPC at Maríubakki must have been transported into the area from surrounding deposits. Comparing dispersed model output with data collected at a single point location is challenging and non-ideal for a model calibration (e.g.

Webster et al., 2012). Possible explanations for the offset in the observed and modelled peak air concentrations could be due to the NWP model not accurately representing the local topography, leading to errors in the modelled wind vectors, or uncertainty in the modelled precipitation. It could also be associated with uncertainty in the defined source areas or uncertainty associated with the OPC data.

The modelled location of the resuspended ash during the 16–17 September 2013 at the times corresponding to the VIIRS data are shown by the blue lines in Fig. 5. The extent of the ash cloud is determined from un-calibrated 1-hour averaged total column mass loadings. Values $> 1 \times 10^{-7}$ g m$^{-2}$ are considered, with this threshold taken as a pragmatic plotting choice to identify the edge of the cloud. Figure 5 shows that ash is resuspended from both the Eyjafjallajökull and Grímsvötn deposits and transported to the south-east over the North Atlantic on the 16 September and then to the south-west as the wind changes

direction on the 17 September (Fig. 2). Both the location and timing of the modelled ash cloud agree well with the VIIRS daytime RGB composites (c.f. Fig. 4). Figure 7 shows the maximum height of the modelled ash cloud and indicates that ash resided at low levels in the atmosphere, $<1600$ m asl on 16 September and $< 2000$ m asl on 17 September. This suggests that the ash cloud was trapped below the temperature inversion, at ∼1500 m (Fig. 2).

### 3.1.1  Brightness Temperature Difference (BTD) Signal

The brightness temperature difference between VIIRS bands M15 and M16, $BTD_V = BT_{M15} - BT_{M16}$, can be used to detect volcanic ash. To determine the expected $BTD_V$ signal for the altitudes at which the resuspended ash cloud resided during the 16–17 September 2013 radiative transfer calculations were carried out for a number of ash cloud top heights. Figure 8a shows calculated $BTD_V$ for a 1 km thick ash cloud with varying ash mass loadings, and ash cloud top heights ranging from 0.5 to





10.0 km. For the ash cloud with a maximum altitude of 0.5 km the ash concentration was increased to resemble that of a 1 km thick cloud. In addition, a simulation with all the ash in a 10 cm thick layer on the surface was included. The assumption of an ash layer with a thickness of 1.0 km is based on the plume heights predicted using NAME (Fig. 7). Ash particles were assumed to have a lognormal size distribution with effective radius $r_e = 2.0$ $\mu$m and geometric standard deviation $\sigma = 2.0$ and nadir

viewing geometry was adopted. It is shown that $BTD_V > 0.0$ when the top of the ash cloud is between 0.5-2.0 km and mass loadings are $\geq 0.02$ g m$^{-2}$. As the ash cloud top height increases $BTD_V$ becomes negative for mass loadings less than 0.05 g m$^{-2}$ for altitudes $< 10$ km. As the 16-17 September 2013 resuspended ash cloud top is between 1-2 km a positive $BTD_V$ signal is therefore to be expected for volcanic ash, as seen in Supplementary Fig. S6.

The absorption of radiation by atmospheric water vapour is larger at 12.0 $\mu$m than at 11.0 $\mu$m. Hence, the presence of water vapour may reduce the volcanic ash $BTD_V$ signal. To remove the water vapour contribution to the $BTD_V$ signal both empirical (Yu et al., 2002) and model based (Corradini et al., 2008; Francis et al., 2012) correction procedures have been developed. Corradini et al. (2008) present the following correction procedure for water vapour absorption:

$$BTD_V^c = BTD_V - BTD_w \qquad (2)$$

where $BTD_w$ is the BTD with water vapour and without ash:

$$BTD_V = T_{15} - T_{16} \qquad (3)$$
$$BTD_w = T_{15}^m - T_{16}^m. \qquad (4)$$

Here $T_{15,16}$ are the measured brightness temperatures in VIIRS bands M15 and M16 respectively, and $T_{15,16}^m$ are the modelled brightness temperatures including only water vapour. Such a correction procedure assumes that radiation from the water

vapour is independent from the radiation from the ash cloud. This assumption may be tested by simulating $BTD_V$ for various ash cloud heights and ash mass loadings with ($BTD_w^{mod}$) and without ($BTD_{w=0.0}^{mod}$) water vapour. The $BTD_{w=0.0}^{mod}$ then resembles $BTD_V^c$ in Eqn. 2, while $BTD_w^{mod}$ resembles $BTD_V$. In view of Eqn. 2, $BTD_w - BTD_{w=0.0}$ should then be constant.

Figure 8b shows the difference $BTD_w^{mod} - BTD_{w=0.0}^{mod}$ for various ash mass loadings as a function of ash cloud top height.

Above an ash cloud top altitude of $\sim$5.0 km the difference becomes constant for all mass loadings. However, the magnitude of the difference decreases with increasing mass loading. Below 5.0 km $BTD_w^{mod} - BTD_{w=0.0}^{mod}$ becomes smaller than the constant value above 5.0 km. The deviation from the constant value increases with increasing ash cloud mass loading. Most of the water vapour is located in the lower troposphere. For an ash cloud above 5.0 km the radiation emitted by the water vapour must traverse the ash cloud similarily to the radiation emitted by the Earth's surface. It will contribute to $BTD_V$ in an additive

manner, c.f. Eqn. 2. For an ash cloud below 5.0 km some of the water vapour will be above and some below the ash cloud. Radiation emitted by the water vapour above the ash cloud does not interact with the ash cloud, hence $BTD_w^{mod} - BTD_{w=0.0}^{mod}$





decreases. For thick ash clouds the water vapour below the ash cloud does not contribute to the signal at the top of the atmosphere.

The 16–17 September 2013 resuspended ash cloud had a top height of about 1.0 km (Fig. 7). As is evident from Fig. 8 any water vapour correction for an ash cloud at this altitude is not straightforward. Hence, we choose to not perform any water vapor correction. Instead we include water vapor based on ECMWF analysis, see Section 2.3 and Supplementary Figs. S1–S4, in the look-up-table calculations needed for the ash mass loading retrieval.

### 3.1.2 Ash pixel detection

Identification of ash pixels can normally be achieved by searching for pixels with $BTD_V < T_{limit}$, where $T_{limit}$ is zero. However, this limit assumes that the ash resides at high altitudes, such that the ash cloud temperature is sufficiently different from the surface temperature (Prata and Grant, 2001). The resuspended ash cloud during the 16–17 September 2013 is easily identifed in the RGB composites (Fig. 4). By comparing the RGB composites with the $BTD_V$ in Fig. S6, the resuspended ash cloud can be clearly identified in both the daytime and night-time images. However, due to the altitude of the resuspended ash cloud during this event $BTD_V > 0.0$ (see Section 3.1.1 and Supplementary Fig. S6) and the normal threshold for identifying ash pixels can not be applied. Instead pixels are identified as containing ash if:

$$(BTD_V > BTD_{min}) \wedge (BTD_V < BTD_{max}) \wedge (BT_{15} > BT_{15_{min}}). \tag{5}$$

The values for $BTD_{min}$, $BTD_{max}$, and $BT_{15_{min}}$ are manually selected upon inspection for each scene and listed in Table 2. The BTD of the pixels identified as containing ash by this procedure is shown in Fig. 9. Through visual inspection of both the daytime (Fig. 4) and night-time images (Supplementary Figs. S5 and S6) areas considered to contain ash are then defined by polygons, as shown in Fig. 4, and Supplementary Figs. S5 and S6, in an attempt to remove the obviously wrongly classified pixels.

### 3.2 Retrieval of ash properties and radiative transfer modelling

From the satellite measurements the ash mass loading may be retrieved. Assuming spherical ash particles the mass loading, $M_l$ (g m$^{-2}$), is given by:

$$M_l = \rho \Delta z_c \int_0^\infty \frac{4}{3} \pi r^3 n(r) dr, \tag{6}$$

where $\rho$ is the density of the ash particles, $\Delta z_c$ is the ash cloud thickness, and $n(r)$ is the ash particle number density distribution. Assuming a log-normal size distribution:



$$n(r) = \frac{N_0}{\sqrt{2\pi}} \frac{1}{\ln(S)} \frac{1}{r} \exp\left[ -\frac{(\ln r - \ln r_0)^2}{2\ln^2(S)} \right], \tag{7}$$

where $N_0$ is the total number of particles per unit volume, $S$ is the geometric standard deviation, and $r_0$ is the geometric mean radius, the mass loading simplifies to:

$$M_l \quad = \quad \rho \Delta z_c \frac{4}{3} \pi N_0 r_e^3 \exp\left( -\frac{6}{2} \ln^2 S \right), \tag{8}$$

where $r_e$ is the ash particle effective radius:

$$r_e \quad = \quad \frac{\int_0^\infty \pi r^3 n(r) dr}{\int_0^\infty \pi r^2 n(r) dr}. \tag{9}$$

It is noted that for the log-normal size distribution, $r_0$ is related to $r_e$ by:

$$r_e = r_0 \exp\left( \frac{5}{2} \ln^2 S \right). \tag{10}$$

It is common to assume values for $S$ and $\rho$. For the case studied here, $\Delta z_c$ is approximately known from temperature profiles and dispersion model calculations. Thus we have:

$$M_l \quad = \quad M_l(N_0, r_e). \tag{11}$$

The VIIRS infrared measurements provides brightness temperatures, $BT$. The brightness temperature is a function of the state of the atmosphere and the underlying surface. This relationship is described by the radiative transfer equation. The state of the atmosphere is described by the temperature profile, the density profiles of relevant trace gases (for example $H_2O$), liquid water and ice cloud particle densities and ash cloud particle densities. For infrared radiative transfer the temperature and emissivity of the underlying surface is also needed. In addition knowledge about the absorption and scattering across sections of the atmospheric constituents is required. For example the ash cloud optical depth $\tau_a$ is given by:

$$\tau_a(\lambda) = \Delta z_c \int Q_{ext}(\lambda, r) \pi r^2 n(r) dr \tag{12}$$

where $Q_{ext}(\lambda, r)$ is the ash cloud extinction efficiency as a function of wavelength $\lambda$ and radius $r$, and a vertically homogeneous ash cloud is assumed.



If we adopt best guess values for the parameters listed in Table 3, the brightness temperature becomes a function of $N_0$ and $r_e$:

$$BT = BT(N_0, r_e). \tag{13}$$

For the ash mass loading estimate we thus tabulate $BT_i$ as a function of $N_0$ and $r_e$ for $i =$M15, M16. The tabulated values

are then used to retrieve $N_0$ and $r_e$ from measured $BT_{M15}$ and $BT_{M16}$ and finally the mass loading is calculated using Eqn. 8.

The retrieval of $N_0$ and $r_e$ is done using the Bayesian method described by Rodgers (2000). The cost function, $J(\mathbf{x})$:

$$J(\mathbf{x}) = (\mathbf{x} - \mathbf{x^b})^{\mathbf{T}} \mathbf{B}^{-1} (\mathbf{x} - \mathbf{x^b}) + (\mathbf{y^{ob}} - \mathbf{y(x)})^{\mathbf{T}} \mathbf{R}^{-1} (\mathbf{y^{ob}} - \mathbf{y(x)}). \tag{14}$$

is minimized using the Levenberg-Marquardt method. Here $\mathbf{x}$ is the atmospheric state vector consisting of the two elements

$(N_0, r_e)$, and $\mathbf{y(x)}$ is the brightness temperature calculated by the forward model for the atmospheric state $\mathbf{x}$, $\mathbf{y}^{ob}$ is the observed brightness temperatures of VIIRS bands M15 and M16. The prior estimate $\mathbf{x}^b$ is set to $(N_0 = 10^6, r_e = 1.0~\mu\text{m})$. The background error covariance matrix is assumed to be diagonal with elements $\sigma^2_{N_0} = (10^{12})^2$ and $\sigma^2_{r_e} = (10\mu\text{m})^2$. The latter value is adopted from Francis et al. (2012). The diagonal elements of $\mathbf{B}$ are large implying that the background state only provides a weak constraint on the retrieved values. The error covariance matrix $\mathbf{R}$ is also assumed to be diagonal. Its diago-

nal elements, $\sigma^2_i$, are the combined variance of the observational and forward model variances. The observational variances are $\sigma^2_{M15} = (0.0028~\text{K})^2$ and $\sigma^2_{M16} = (0.0036~\text{K})^2$ and the forward model variance taken as $\sigma^2_{FM} = (1.0~\text{K})^2$. This gives $\sigma^2_i = (1.0~\text{K})^2$.

The uvspec tool from the libRadtran radiative transfer package (Mayer and Kylling, 2005; Emde et al., 2011, and www.

libradtran.org) was used as the forward model to calculate VIIRS brightness temperatures for bands M15 and M16. A plane-parallel atmosphere was assumed and the discrete-ordinate method was used to solve the radiative transfer equation with 16 streams (Stamnes et al., 1988; Buras et al., 2011). The ambient atmosphere profiles of temperature, pressure and water vapour were taken from the averaged ECMWF profiles as described in Section 2.3 (Supplementary Figs. S1–S4). The surface was assumed to be sea water with wavelength emissivity taken from http://www.icess.ucsb.edu/modis/EMIS/html/seawater.html.

For the gas absorption the REPTRAN parameterization was used (Gasteiger et al., 2014). The resuspended ash was included as a plane-parallel layer. The ash particles were taken to be of andesite composition and the refractive index was adopted from Pollack et al. (1973). The ash particles were assumed to be spherical in shape and their optical properties were calculated using Mie theory. It is noted that porosity and non-sphericity of the ash particles may affect the electromagnetic IR radiation measured by the VIIRS (Kylling et al., 2014). The uvspec model is computationally too slow to be used on-line in the retrieval

therefore look-up-tables (LUT) were calculated as a function of $N_0$ and $r_e$ for surface temperatures between 280-284 K. Figure 10 shows the retrieved ash mass loading of the resuspended ash cloud for the areas identified as containing ash, and Table 2





gives the total mass retrieved. The location of ash cloud agrees well with the forecasts using NAME (c.f. Fig. 5). Quantifying the uncertainty on satellite retrievals of volcanic ash is non-trivial, and includes uncertainties in the retrieval and uncertainties in the assumed parameters such as refractive index and particle size distribution. Based on the work by Corradini et al. (2008) and in addition considering the uncertainty due to particle shape (Kylling et al., 2014) we assign an uncertainty of $\pm 50\%$ to

the total mass retrieved for each image.

## 4   Quantifying the total mass of ash resuspended

Here we determine the scaling coefficient ($K$) for the emission rate ($F$, Eqn. 1) of remobilized ash in NAME. As we have data from only one OPC instrument we are unable to perform a robust calibration with surface $PM_{10}$ data. Instead we calibrate the

emission rate in NAME such that modelled total column mass loadings match those retrieved from VIIRS.

A peak-to-peak calibration can be determined by calculating the difference between the mode of the VIIRS retrieved total column mass loadings (contained within the polygons) and the mode of the un-calibrated NAME output. Figure 11 shows the frequency of binned total column mass loadings from the satellite retrievals and the un-calibrated model output for each

retrieval time. The mode of the VIIRS mass loadings varies with time during the event, from $10^{-1} - 10^{0}$ g m$^{-2}$ to $10^{0} - 10^{1}$ g m$^{-2}$, this variation includes the uncertainty associated with the retrieval. The un-calibrated modelled total column mass loadings have a mode at $10^{-4} - 10^{-3}$ g m$^{-2}$. Considering the difference in the mode of the VIIRS retrieved mass loadings and the model output at each retrieval time suggests we need to apply a calibration factor of between $K = 1 \times 10^{3} - 1 \times 10^{4}$ to the resuspension scheme in NAME to match the observed mass loadings in the atmosphere.

Simulated mass loadings using these calibration factors are given in Fig. 12. The performance of the calibration factors are assessed by calculating the Fractional Bias between the satellite retrieved and the modelled total column mass loadings within the polygons (Table 4). The Fractional Bias is a measure of the mean bias and indicates over or under-estimation of the model output, values range between -2 and +2, a positive value represents over-prediction of the model with respect to the VIIRS

retrieved mass loadings and a negative value under-prediction, a value of 0 represents a perfect match. Scaling the source strength by $K = 1 \times 10^{4}$ systematically overestimates mass loadings, whereas using $K = 1 \times 10^{3}$ results in a better match to the satellite retrievals. This is still the case when we consider that the retrieved mass loadings have an uncertainty of $\pm 50\%$. Using a source strength scaled by $K = 1 \times 10^{3}$ the total mass of ash resuspended from the model Eyjafjallajökull 2010 and Grímsvötn 2011 deposits between 00:00 UTC on the 16 September 2013 to 00:00 UTC on 18 September 2013 is $\sim 0.2$ Tg.





## 5 Discussion

The total mass of ash erupted from Eyjafjallajökull in 2010 was estimated from ground surveys and remote sensing to be $384 \pm 96$ Tg (Gudmundsson et al., 2012). Calculated estimates using plume rise models are also found to lie within the error bounds of this observational estimate (Devenish, 2016). Preliminary results from mapping the Grímsvötn 2011 fall deposits indicate that the bulk volume of ash from this eruption is two to three times larger (Gudmundsson et al., 2012). We estimate that $\sim 0.2$ Tg of ash was remobilised during 16–17 September 2013.

The calibration applied in this study is uniquely related to the event studied and the source areas defined. Our methodology does not account for how the deposits and rates of resuspension may vary over time. The calibrated emission flux of $K = 1 \times 10^3$ is lower than the original calibration determined by Leadbetter et al. (2012), $K = 1.1 \times 10^7$ (taking an emission flux in grams) for the Eyjafjallajökull ash source in 2010. This suggests that resuspension rates had declined by 2013, perhaps due to depletion and compaction of the ash with time since it was deposited or re-growth of vegetation. However, the retrieved mass loadings from VIIRS and the calibrated modelled mass loadings show that the resuspended ash cloud contained significant quantities of ash. Dividing the calculated total mass of ash resuspended over the emission time period (48 hours) we calculate an average emission rate of $1.04 \times 10^3$ kg s$^{-1}$. This is equivalent to the minimum calculated eruption rates of tephra from Eyjafjallajökull 2010 using plume rise models, which range between $10^3$ – $10^6$ kg s$^{-1}$ over the 39 day eruption (Devenish, 2013; Woodhouse et al., 2012). The magnitude of the retrieved ash mass loading in individual scenes from the VIIRS data is also comparable in magnitude to those determined by Prata and Prata (2012) using SEVIRI of the distal ash cloud from the eruption of Eyjafjallajökulll 2010 over the southern North Sea on the 17 May 2010. This suggests that remobilisation of ash deposits can produce ash clouds with mass loadings equivalent to those observed from explosive volcanic eruptions. One important distinction is that the buoyant ash plume generated from the eruption of Eyjafjallajökull released ash to altitudes up to 10 km asl, and the resulting ash cloud was consequently transported by upper air winds. Whereas resuspended ash, remobilised from deposits, is necessarily closer to the surface, and during the 16–17 September 2013 the ash was trapped below a temperature inversion at $< 2$ km asl restricting further vertical dispersion. Ash sedimenting from a low altitude resuspended ash cloud will be deposited quicker than ash which is released at upper levels, as it does not have as far to fall and because it will be rained-out by precipitation from clouds formed above the ash layer.

Here we have considered particles with diameter $\leq 10$ $\mu$m only, to be consistent with the particle size range which can be detected by the satellite retrievals. However Liu et al. (2014) showed that much larger grains, up to 177 $\mu$m in diameter, can be resuspended from the Eyjafjallajökull and Grímsvötn deposits and transported significant distances. This would suggest that the resuspended ash cloud could have included much larger particles that have not been accounted for in our scaled emission flux, as such our calculated total mass loadings could represent a minimum estimate.





We have used the extent of tephra deposits defined immediately after the eruptions of Eyjafjallajökull in 2010 and Grímsvötn in 2011 to identify the potential source area from which ash can be resuspended. This does not consider how the deposits may have been modifed since they were formed. Compaction and cementation processes increase deposit cohesion and can reduce the emission flux of particles. Here, we have applied the same scaling coefficient to both the Eyjafjallajökull and Grímsvötn

deposits, which could under-estimate the flux from the younger Grímsvötn deposits and over-estimate the flux of particles from the older Eyjafjallajökull deposits (Liu et al., 2014). Deposits are also re-distributed as ash is resuspended, advected and re-deposited. Jökulhlaups (sub-glacial floods) can also transport large volumes of ash which is then re-deposited on outwash planes (sandurs). The sandur planes represent large areas of unstable sediments, and are known to be an additional source of remobilised particles across Iceland (Arnalds et al., 2001, 2014; Dagsson-Waldhauserova et al., 2014; Arnalds et al., 2016).

Arnalds et al. (2014) calculated the total emission from a remobilised 'dust storm' on the 25 May 2012 in Dyngjusandur, a large glacio-fluvial plain north of Vatnajökull, to be $3.65 \times 10^5$ tons ($\sim$0.3 Tg). The calculated emission is based on measurements of the horizontal extent of the plume and visibility (weather) observations, which were validated with MODIS satellite imagery. More recently Dagsson-Waldhauserova et al. (2016) estimated the total mass of dust resuspended during two storms in south-west Iceland, on the 15 June 2015 and the 4 August 2015, from observations of the horizontal extent of the plume and

visibility measurements to be $\sim$0.18 Tg and $\sim$0.28 Tg respectively. The VIIRS satellite imagery of the resuspended ash cloud during the 16–17 September 2013 clearly indicates that the source of the remoblised ash cloud is over southern Iceland, and the two distinct plumes observed in the visible imagery (Fig. 4) suggest that both the Eyjafjallajökull and Grímsvötn deposits are the primary sources of the remobilised ash. The good agreement between the modelled and observed location and timing of the resuspended ash cloud gives us confidence that our source areas are well defined (Fig. 5). However, the sandur planes

on the south coast at Mýrdalssandur and Skeiðarársandur may also have been an additional source of ash which has not been accounted for. It is not yet understood whether the mechanism of resuspension, and hence the rate at which particles are remobilised, from the sandur planes differs to that from the tephra deposits. Applying the same calibration coefficient to a larger source area, to include the sandur plains, would increase the total modelled emission flux.

## 6   Conclusions

Volcanic ash continues to pose a hazard to local populations and airports for years after an eruption as particles are remobilised from deposits. NAME, which includes a resuspension scheme for volcanic ash, is used to provide daily forecasts of possible remobilised ash storms in Iceland. When a significant resuspension event is anticipated the local population is informed by the IMO via their routine weather forecasts. To forecast resuspended ash storms with dispersion models the source (deposit)

areas and the emission rate of the particles must be known. This is challenging because deposits continuously evolve as they are remobilised, compacted and revegetated. Here we have applied a novel technique to constrain the emission rate in the resuspension scheme in NAME using retrieved mass loadings of a resuspended ash cloud from satellite imagery. The simple approach presented here, in which the source strength is scaled by a calibration factor ($K$) to observations is very versatile. It





allows the user to update the emission scheme with time, matching to observations as deposits evolve. We find that a calibration factor of $K = 1 \times 10^3$ best represents ash mass loadings of a resuspended ash cloud observed during the 16–17 September 2013 over southern Iceland. Using this calibration factor we estimate that a total of $\sim$0.2 Tg of ash was remobilised during this event.

## 5   Data availability

VIIRS data are available from the NASA VIIRS Atmosphere SIPS (sips.ssec.wisc.edu/, NASA, 2016). NAME, meteorological data, processed data, analysis results, and analysis and visualization codes are available upon request from the Met Office (email to atmospheric.dispersion@metoffice.gov.uk).

### Acknowledgments

10   FMB would like to thank Ayoe Hansen (ADAQ, Met Office) for her help in plotting the tephigrams, Michael Lawrence (EMARC Meteorologist, Met Office) for his advise on interpreting them and Susan Leadbetter and Matthew Hort (ADAQ, Met Office) for their reviews of a draft of this manuscript. AK acknowledges support from the Norwegian Research Council (Contract 224716/E10). Funding for this research was provided by the FP7 project FUTUREVOLC 'A European volcanological supersite in Iceland: a monitoring system and network for the future', (Grant agreement no: 308377).

### Author Contributions

Frances Beckett performed the NAME simulations and wrote most of the manuscruipt. Arve Kylling produced the satellite imagery, performed the satellite retrievals and the radiative transfer modelling, and provided significant input to writing the manuscript. Guðmunda Sigurðardóttir and Sibylle von Löwis provided the OPC data and advice on intepreting it. Claire 20   Witham provided advice and guidance on the NAME modelling and calibration of the source strength.



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





**Table 1.** VIIRS bands used in this study. Further information is available from https://cs.star.nesdis.noaa.gov/pub/NCC/UsersGuideVIIRS/
VIIRS_USERS_GUIDE_Tech_Report_142A_v1.2.pdf

| Band no | Central wavelength ($\mu$m) | Usage |
|---|---|---|
| M3 | 0.488 | RGB composite |
| M4 | 0.555 | RGB composite |
| M5 | 0.672 | RGB composite |
| M15 | 10.763 | Ash detection and retrieval |
| M16 | 12.013 | Ash detection and retrieval |

**Table 2.** VIIRS data used in this study. The study area is limited to the area delimited by 65°N, 26°W, 12°W, 54°N. VIIRS data were ordered
from http://www.nsof.class.noaa.gov/saa/products/.

| Date | Time (UTC) (start of scan) | $BTD_{min}$ (K) | $BTD_{max}$ (K) | $BT_{M15_{min}}$ (K) | Total ash mass (Gg) |
|---|---|---|---|---|---|
| 16/09/2013 | 02:06 | 0.0 | 0.8 | 272.0 | 17.78 |
| 16/09/2013 | 03:42 | 0.0 | 0.5 | 275.0 | 11.80 |
| 16/09/2013 | 05:24 | 0.0 | 0.8 | 270.0 | 17.05 |
| 16/09/2013 | 12:00 | 0.0 | 0.8 | 270.0 | 19.52 |
| 16/09/2013 | 13:36 | -0.1 | 0.45 | 270.0 | 14.63 |
| 16/09/2013 | 15:18 | -0.1 | 1.0 | 270.0 | 24.89 |
| 17/09/2013 | 03:24 | -0.1 | 0.7 | 275.0 | 26.58 |
| 17/09/2013 | 05:06 | -0.1 | 0.8 | 275.0 | 8.75 |
| 17/09/2013 | 11:42 | 0.3 | 1.0 | 275.0 | 13.76 |
| 17/09/2013 | 13:18 | 0.0 | 0.5 | 275.0 | 8.67 |
| 17/09/2013 | 15:00 | 0.0 | 1.0 | 275.0 | 8.05 |





**Table 3.** Assumed parameters and values used for the ash cloud retrieval. 'ECMWF average' means the parameter is calculated from ECMWF analysis data averaged over the region for 16-17 September 2013. See text for more details.

| Parameter | Value | Comment |
|---|---|---|
| $\rho$ | 2300 (kg m$^{-3}$) | Ash particle density |
| $T_c$ | ECMWF average | Temperature of ash cloud top, Supplementary Fig. S4. |
| $\Delta z_c$ | 1000.0 m | Ash cloud thickness |
| $n(r)$ | log-normal | Particle number density distribution |
| $S$ | 2.0 | Geometric standard deviation |
| $T_s$ | ECMWF average | Surface temperature, Supplementary Fig. S3. |
| $\epsilon$ | Sea water | Emissivity of surface |
| $T(z)$ | ECMWF average | Temperature profile, Supplementary Fig. S4. |
| $Q_{ext}$ | Andesite | Ash type |
| $\rho_{H_2O}(z)$ | ECMWF average | Water vapour profile, Supplementary Fig. S2. |

**Table 4.** Calculated Fractional Bias between VIIRS retrieved total column mass loadings and modelled total column mass loadings where the emission rate in NAME is calibrated using the scaling coefficient ($K$) derived from a peak to peak scaling to the VIIRS data.

| Time + Date | $K = 1 \times 10^3$ | $K = 1 \times 10^4$ |
|---|---|---|
| 02:06 16/09/2013 | 0.89 | 1.85 |
| 03:42 16/09/2013 | 1.14 | 1.89 |
| 05:24 16/09/2013 | 0.22 | 1.71 |
| 12:00 16/09/2013 | 0.40 | 1.75 |
| 13:36 16/09/2013 | 0.51 | 1.78 |
| 15:18 16/09/2013 | 0.70 | 1.82 |
| 03:24 17/09/2013 | -0.61 | 1.37 |
| 05:06 17/09/2013 | 0.82 | 1.84 |
| 11:42 17/09/2013 | 0.40 | 1.75 |
| 13:18 17/09/2013 | 0.77 | 1.83 |
| 15:00 17/09/2013 | 1.16 | 1.90 |



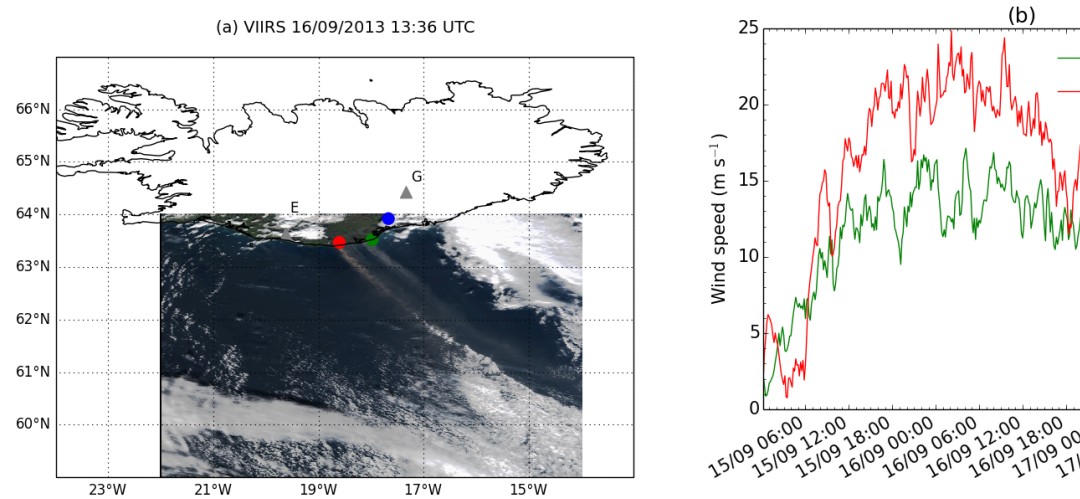

**Figure 1.** (a) "True" colour VIIRS daytime image for 13:35 UTC on the 16 September 2013 with the locations of the Skarðsfjöruviti (green marker) and Mýrdalssandur (red marker) weather stations. The location of the OPC, at Maruibakki is indicated by the blue marker. The locations of the volcanoes Eyjafjallajökull and Grímsvötn are indicated on the map by the E and G symbols respectively. (b) The recorded wind speeds at Skarðsfjöruviti and Mýrdalssandur during the 15–17 September 2013.




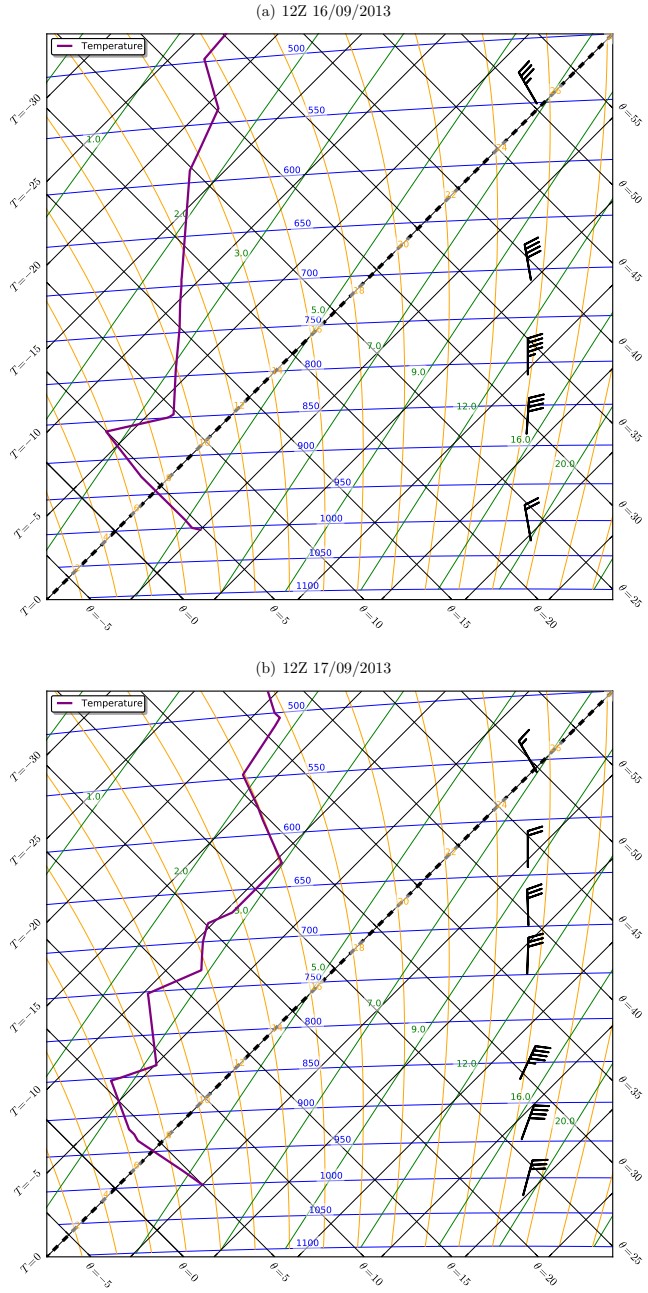

**Figure 2.** Radiosonde data retrived from launches at Keflavík airport at 12:00 UTC on (a) the 16 and (b) 17 September 2013. North-westerly surface winds prevail on the 16 September and a temperature inversion is observed at 850 hPa, $\sim$ 1500 m asl (where pressure is indicated by the blue lines). On 17 September surface winds have veered north-easterly, the temperature inversion remains although it is now weaker. Radiosonde data were obtained from http://weather.uwyo.edu/upperair/sounding.html



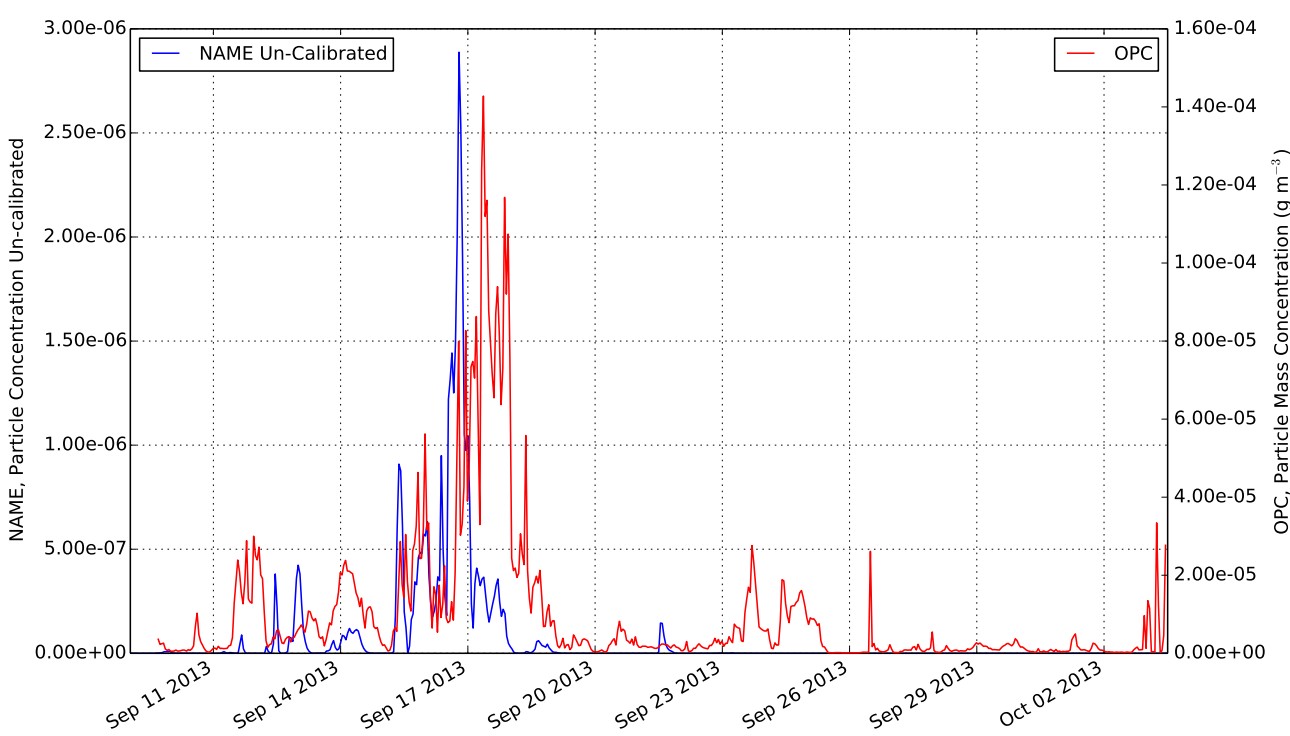

**Figure 3.** Time series of 1 hour averaged concentrations of resuspended ash derived from OPC count data (red) compared to the un-calibrated modelled air concentrations (blue) at Maríubakki during September 2013.



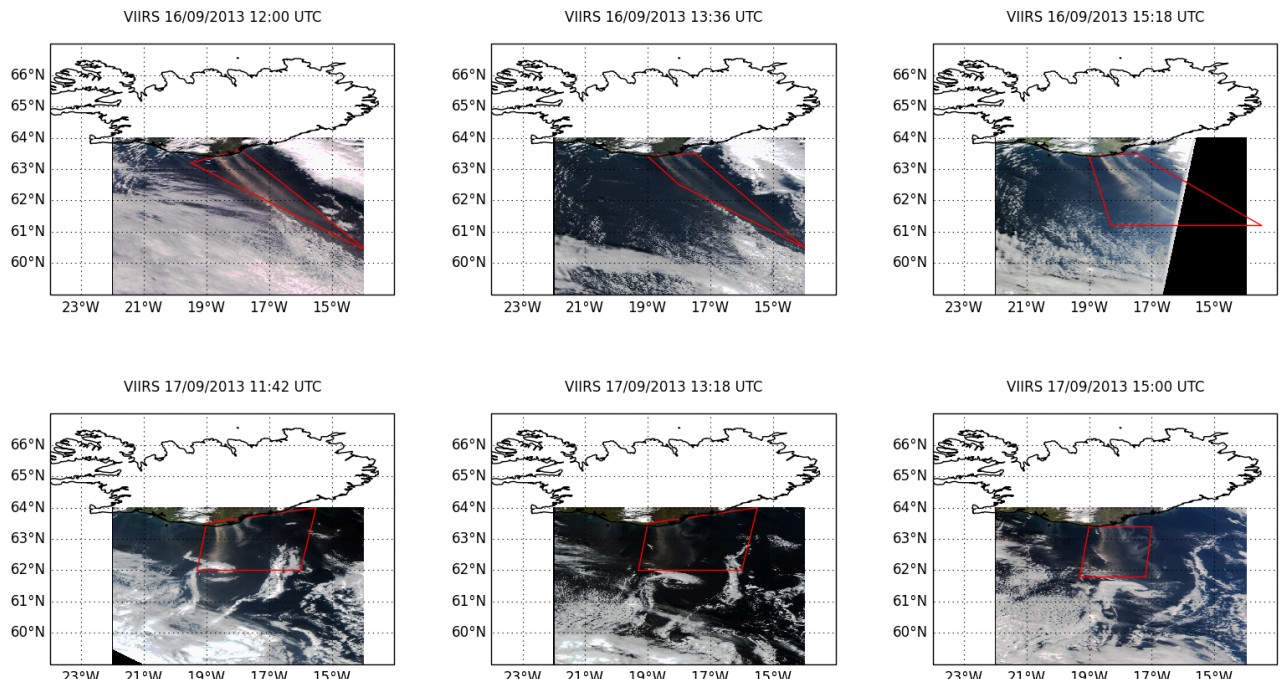

**Figure 4.** VIIRS daytime RGB composites using bands M3, M4, and M5, for 16-17 September 2013. Areas identified as containing resuspended ash, see Section 3.1.2, are enclosed within the red polygons.




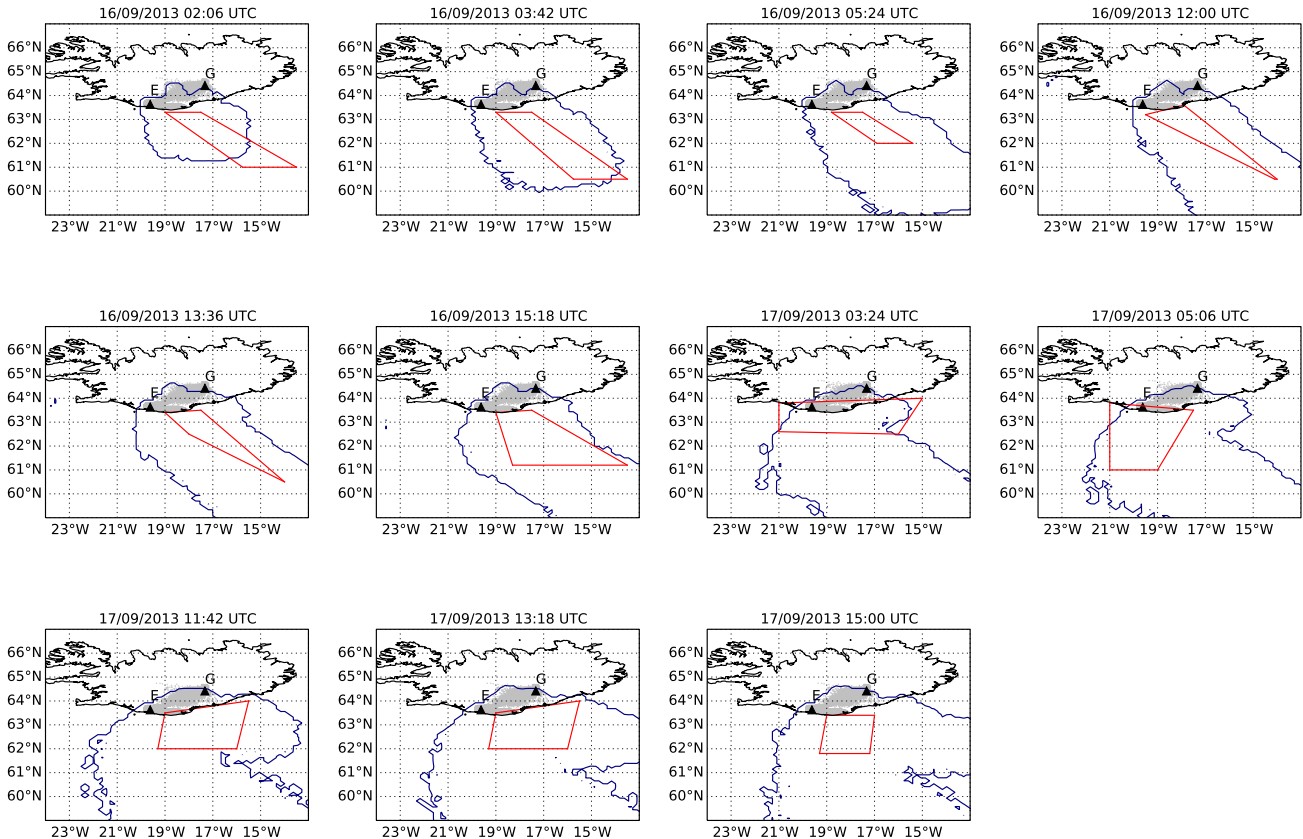

**Figure 5.** To compare the location of the modelled resuspended ash cloud, represented by the blue line, to the area identified from VIIRS retrieval data, represented by the red polygons, see Section 3.1.2 for the methodology used to define this area. The outline of the modelled plume is derived from un-calibrated 1 hour averaged total column mass loadings, values $> 10^{-7}$ gm$^{-2}$ are considered. The source areas are identified by the grey areas.



**Figure 6.** Time series of 1 hour averaged air concentrations of resuspended ash derived from OPC count data and the un-calibrated model output, compared to the NAE precipitation rate and friction velocity ($U*$) at Maríubakki.





**Figure 7.** The maximum height of the modelled ash cloud using NAME at the times corresponding to the satellite retrievals. The locations of the volcanoes Eyjafjallajökull and Grímsvötn are indicated on the map by the E and G symbols respectively.





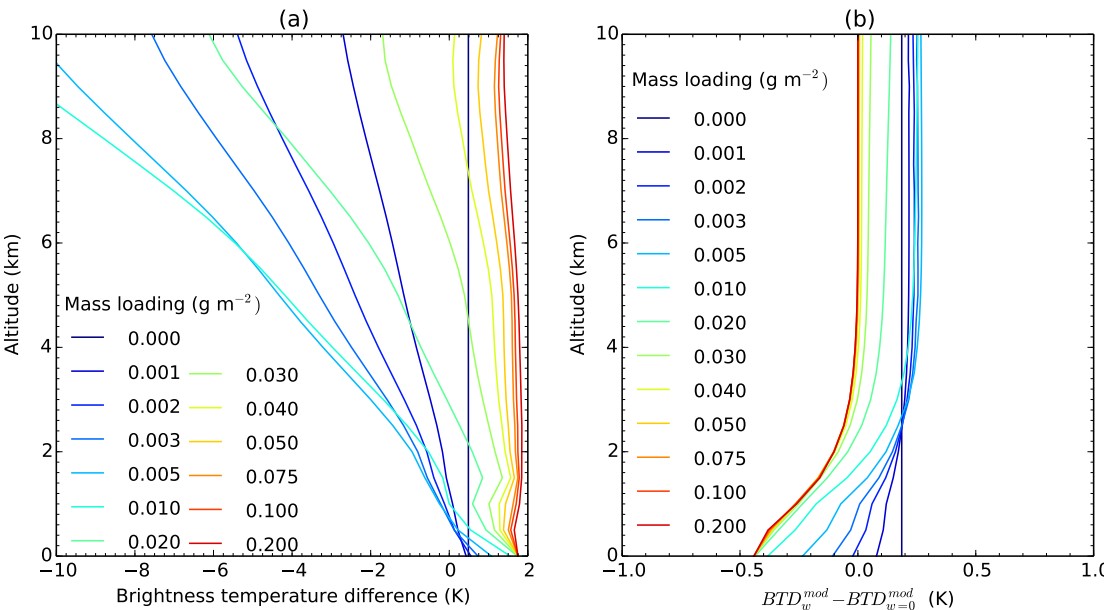

**Figure 8.** (a) The simulated brightness temperature difference between VIIRS bands M15 and M16, $BTD_V$, for a 1 km thick ash layer as function of ash layer top altitude. (b) The difference $BTD_w^{mod} - BTD_{w=0.0}^{mod}$ as a function of ash cloud top height. The curves represent varying ash mass loading (g m$^{-2}$) and are given in the legend.





**Figure 9.** VIIRS BTDs for pixels identified as resuspended ash, 16-17 September 2013.





**Figure 10.** The resuspended ash mass loading retrieved from VIIRS infrared bands M15 and M16 for the areas identifed as containing ash.





**Figure 11.** Comparing the frequency of binned total column mass loadings of the resuspended ash cloud modelled using NAME with an uncalibrated source strength to those retrieved from VIIRS during the 16–17 September 2013.





(a)

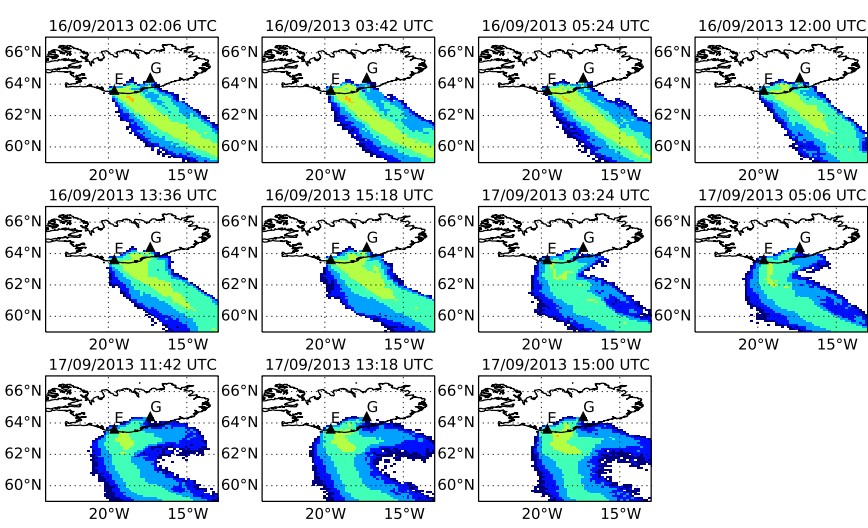

(b)

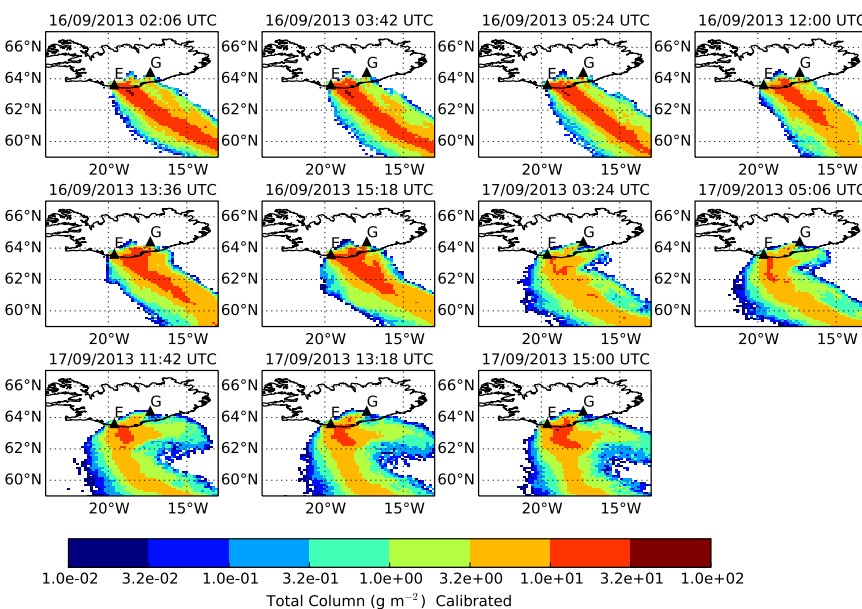

**Figure 12.** Modelled 1 hour averaged total column mass loadings $(\mathrm{g\,m^{-2}})$, where the source strength in NAME is calibrated using the scaling coefficient determined from the peak to peak scaling to the satellite retrieved total column values (a) $K = 1 \times 10^{3}$ and (b) $K = 1 \times 10^{4}$. The locations of the volcanoes Eyjafjallajökull and Grímsvötn are indicated on the map by the E and G symbols respectively.