# Peer review of "Quantifying the mass loading of particles in an ash cloud remobilised from tephra deposits on Iceland"

_Atmospheric Chemistry and Physics, 2016_

## Referee Comment (RC1) · Anonymous Referee #1 · 2 Nov 2016

This is an interesting and very well written and presented paper that I think is suitable for publication. My comments are limited to very minor clarifications and further questions that the authors may choose to expand upon. The authors use VIIRS measurements to quantify the mass of remobilised Icelandic tephra at low altitude over the North Atlantic on two days in September 2013. They demonstrate (1) the use of a positive brightness signal for identification of ash in VIIRS data (by comparing the observed distribution to NAME model outputs) and (2) the quantification of the mass of re-suspended ash using the VIRR column mass loadings to calibrate the scaling coefficient for the emission (re-suspension) rate.

- This article is an interesting demonstration of a method for quantifying the mass of

low altitude ash, and it would be useful to have some additional comments from the authors on general applicability. As pointed out in the 2nd paragraph of the discussion, the emission rate calibration for NAME is case specific – but could a similar approach (calibration from VIIRS for a particular date) be applied to track the dropping re-suspension rate since eruptions in Iceland?

- Last sentence of abstract: on first reading this confused me – would help to clarify here that this refers to assumed source area for those eruptions.

- Introduction. It would be useful to have a bit more information about how common re-suspension events are?

- Section 3.1.2. Could this 'positive BTD signal' approach be applied more generally? To what extent is it limited to specific meteorological conditions/height of emission - e.g., is there an ash cloud height and mass loading for which BTD is too close to zero to be useful? From Figure S6 it looks like BTDmin must have been very close to zero in some cases.

- Discussion: It's fascinating that a potentially low estimate of resuspended tephra reaches similar mass loadings to Eyjafjallajökull ash on quieter days. I think that the positive BTD approach might be interesting for volcanologists tracking ash emission from frequent, but lower-explosivity eruptions (e.g., VEI 2 or 3 events or even frequent vulcanian events that emit ash at ∼few km elevation). Even without being able to estimate total loading from a calibrated model, this could provide volcanologically useful information.

---

## Referee Comment (RC2) · M. Watson (Referee) · 15 Dec 2016

Review of Beckett et al., ACP. Matthew Watson, Bristol, December 2016

Overview: This paper details satellite observations (VIIRS) and dispersion modelling (NAME) of re-suspended volcanic ash from Iceland during an event in September 2013. It is well written, with very few editorial issues, appears well referenced and does, I think, provide incremental improvement in our understanding of the phenomenon. There are however some serious issues with the paper which will require revisions before the paper can be published.

Major: The per-pixel mass loadings from VIIRS can be worked up into total mass. This

is not discussed in the text in although appears to have been done (in Table 2, which is also only fleetingly discussed). Are those numbers (on order 10 Gg) comparable with the NAME estimates of 0.2 Tg? Why is there a five order of magnitude difference? It doesn't appear that you can even get to 10 Gg from summing the column loadings in figure 10. (A quick back of envelope calculations using an area 2E9 m2 (much larger than the observed clouds) and the max observed loading gives 6E9 g (i.e. 6 Tg)? This may simple be a typo, but needs resolving.

Following from that, the whole process appears rather circular. It's a somewhat tortuous process to go from column loading, to emission rate, to mass from the NAME model when the VIIRS observations tell you that directly? This entire section is quite confusing (section 4), for example it's not clear what 'un-calibrated really means'. NAME must have been run with some starting conditions with some given units (even if this is unity). Also, the use of the scaling factor is poorly defined (and has a significant impact on the final mass). It appears to be derived from the difference in masses between the column burdens derived from VIIRS and the uncalibrated NAME runs. This requires significant expansion.

The section on water vapour, whilst technically correct, is completely undermined by the final section of 3.1.1 where the discussion grinds to the halt as it is explained that the water vapour correction was not applied. Recast this section to explain what was done (in more detail) rather than a more complex explanation of something that wasn't.

Why does the BTD signal in Figure 8 get stronger at lower mass (for constant particle size). That is opposite to what every paper I've ever read on the subject would suggest.

There may be other reasons for positive BTD. The authors should probably approach them, and rule them out (especially coating of the ash and/or mixing with ice). I accept that this is unlikely but there are precedents in Iceland, though not from re-suspended ash.

Leadbetter et al., 2012 proposed a range of 0.4 – 0.5 for U*t. What difference would

using 0.5 make (i.e. how sensitive to the final outcome is that choice)?

What effect does limiting the NAME 1-10 microns have? Given much larger (and more mass bearing) particles have been observed (and discussed later in the paper) this is something the authors will need to further explore.

In summary there are some quite unfathomable things in the paper. I would encourage the authors to work through these and provide explanations. It could be I've simply misunderstood but even that would imply a lack of clarity in the paper.

Editorial (very minor): P1 L19 should be ':' not ';' P2 L10 'random walk' might need further explanation / reference P4 L1-5. Reasons for mass loadings are presented but do not mention reduced availability. This is then discussed later in the paper, foreshadow that discussion, briefly here (it seems to be to be a perfectly reasonable explanation, as, of course, does the location of the OPC) P4 L19 Maybe quote the calculate dBTD cost from water vapour here (for the purists). P7 L1 This is clumsy. Do you mean your doubled the concentration of the lowest layer to preserve constant mass? P11 L3 cite 'Mackie, S., Millington, S. and Watson, I.M., 2014. How assumed composition affects the interpretation of satellite observations of volcanic ash. Meteorological Applications, 21(1), pp.20-29.'? P12 L16 Chronologise reference list Figures look good in colour but are unusable in black and white (no change required unless the paper won't be published in colour)

---

## Author Comment (AC1) · 25 Jan 2017

**Response to Reviewer Comments**

**Quantifying the mass loading of particles in an ash cloud remobilised from tephra deposits on Iceland**

Frances Beckett, Arve Kylling, Guðmunda Sigurðardóttir, Sibylle von Löwis and Claire Witham

We thank the referees for their thorough reviews and helpful suggestions. Please find below responses to each comment, provided in blue.

**Anonymous Referee #1**

This is an interesting and very well written and presented paper that I think is suitable for publication. My comments are limited to very minor clarifications and further questions that the authors may choose to expand upon. The authors use VIIRS measurements to quantify the mass of remobilised Icelandic tephra at low altitude over the North Atlantic on two days in September 2013. They demonstrate (1) the use of a positive brightness signal for identification of ash in VIIRS data (by comparing the observed distribution to NAME model outputs) and (2) the quantification of the mass of re-suspended ash using the VIRR column mass loadings to calibrate the scaling coefficient for the emission (re-suspension) rate.

- This article is an interesting demonstration of a method for quantifying the mass of low altitude ash, and it would be useful to have some additional comments from the authors on general applicability. As pointed out in the 2nd paragraph of the discussion, the emission rate calibration for NAME is case specific – but could a similar approach (calibration from VIIRS for a particular date) be applied to track the dropping re-suspension rate since eruptions in Iceland?

Response: We thank the reviewer for this point and now include this in the Introduction and Discussion. Page 3, Line 3 in the Introduction reads:

*'The newly calibrated scheme can be used to provide more accurate quantitative forecasts of future events, and assess how resuspension rates are varying over time.'*

In the Discussion, Page 13, Line 17 reads:

*'The calibration applied in this study is uniquely related to the event studied and the source areas defined, but this approach could be used to consider how the emission rate of resuspension is varying with time since the ash was deposited.'*

- Last sentence of abstract: on first reading this confused me – would help to clarify here that this refers to assumed source area for those eruptions.

Response: We have re-worded the sentence to read:

*'Considering the tephra deposits from the recent eruptions of Eyjafjallajökull and Grímsvötn as the potential source area for resuspension for this event, we estimate that ~0.2 Tg of ash was remobilised during 16-17 September 2013.'*

- Introduction. It would be useful to have a bit more information about how common re-suspension events are?

Response: The following information has been added to the Introduction on Page 2, Line 1:

*'Between the 19 September 2010 and the 16 February 2011 there were 12 observed resuspension episodes recorded by $PM_{10}$ counters in Drangshildardalur (southern Iceland) of the Eyjafjallajökull ash deposits.'*

- Section 3.1.2. Could this 'positive BTD signal' approach be applied more generally? To what extent is it limited to specific meteorological conditions/height of emission - e.g., is there an ash cloud height and mass loading for which BTD is too close to zero to be useful? From Figure S6 it looks like BTDmin must have been very close to zero in some cases.

Response: The approach is adapted to the time, location and situation being studied. The positive BTD signal will depend on the underlying surface. We are over water which, compared to land surfaces, have far less horizontal and temporal variations in the refractive index. The amount of water vapour in the atmosphere will also affect the signal as will the altitude of the resuspended ash. Finally, cloud free pixels are required. We have added the following text to the end of section 3.3:

*'As discussed in the above section, the BTD signal depends on the atmospheric water vapour content, the resuspended ash height and requires cloud free pixels. In addition the optical properties of the underlying surface must be accounted for. The detection method has potential for application in other cases, but must be adapted to the situation being studied.'*

- Discussion: It's fascinating that a potentially low estimate of resuspended tephra reaches similar mass loadings to Eyjafjallajökull ash on quieter days. I think that the positive BTD approach might be interesting for volcanologists tracking ash emission from frequent, but lower-explosivity eruptions (e.g., VEI 2 or 3 events or even frequent vulcanian events that emit ash at ~ few km elevation). Even without being able to estimate total loading from a calibrated model, this could provide volcanologically useful information.

See response above.

**Review of Beckett et al., ACP. Matthew Watson, Bristol**

Overview: This paper details satellite observations (VIIRS) and dispersion modelling (NAME) of re-suspended volcanic ash from Iceland during an event in September 2013. It is well written, with very few editorial issues, appears well referenced and does, I think, provide incremental improvement in our understanding of the phenomenon. There are however some serious issues with the paper which will require revisions before the paper can be published.

Major:

The per-pixel mass loadings from VIIRS can be worked up into total mass. This is not discussed in the text in although appears to have been done (in Table 2, which is also only fleetingly discussed). Are those numbers (on order 10 Gg) comparable with the NAME estimates of 0.2 Tg? Why is there a five order of magnitude difference? It doesn't appear that you can even get to 10 Gg from summing the column loadings in figure 10. (A quick back of envelope calculations using an area 2E9 m2 (much larger than the observed clouds) and the max observed loading gives 6E9 g (i.e. 6 Tg)? This may simple be a typo, but needs resolving.

We thank the reviewer for their back of envelope approach. This demonstrates a total ash mass of 6 Gg (where noting that a Tg is 1E12 g and a Gg is 1E9 g), which is equivalent to the numbers in Table 2. The area of 2E9 $m^2$ chosen by the reviewer is ~ 0.2 deg x 1 deg which is entirely appropriate for the smaller polygons identified in Fig. 9.

The total remobilised ash mass of 0.2 Tg was determined by integrating the calibrated emission rate over the 16-17 September 2013 from the Eyjafjallajökull 2010 and Grímsvötn 2011 deposits. The VIIRS retrieved total mass loadings in Table 2 represent a snap-shot in time, the mass of ash in the atmosphere at the time of the retrieval. If we take a simple approach and assume that the mass retrieved from each over-pass are independent from one another then when you sum the total ash mass from each of the retrievals in Table 2 you achieve a total remobilised mass of 0.171 Tg.

This discussion is now included in the text and we make a clearer reference to the calculated mass loadings from the VIIRS retrievals in Table 2.

In Section 3.4 we have expanded our reference to Table 2, now stating on Page 11 Line 29:

*'Table 2 gives the retrieved mass of ash in the atmosphere for each overpass'*

And at the end of Section 4 we have expanded our discussion on our calculated total mass of ash remobilised during the entire event:

*'Summing the mass loadings from each VIIRS retrieval (Table 2) gives the total observed mass of remobilised ash to be 0.17 Tg. This represents contributions only from the mass in the atmosphere at the time of each overpass and may double-count between retrievals. Using the modelled emission rate, scaled by K = 1 x 10$^3$, the total mass of ash remobilised from the Eyjafjallajökull 2010 and Grímsvötn 2011 deposits between 00:00 UTC on the 16 September 2013 to 00:00 UTC on 18 September 2013 is ~0.2 Tg.'*

Following from that, the whole process appears rather circular. It's a somewhat tortuous process to go from column loading, to emission rate, to mass from the NAME model when the VIIRS observations tell you that directly? This entire section is quite confusing (section 4), for example it's not clear what 'un-calibrated really means'. NAME must have been run with some starting

conditions with some given units (even if this is unity). Also, the use of the scaling factor is poorly defined (and has a significant impact on the final mass). It appears to be derived from the difference in masses between the column burdens derived from VIIRS and the uncalibrated NAME runs. This requires significant expansion.

Response: In this study we have calibrated the emission rate in the resuspension scheme in NAME. This is achieved by applying a scaling coefficient ($K$). $K$ is determined by scaling the mode of NAME modelled mass loadings, where $K$ is initially set to unity, to the mode of the VIIRS retrieved mass loadings. The aim is to enable us to produce more accurate quantitative forecasts for future events (where there are no observations). The satellite retrievals provide a snap-shot in time and allow us to assess the mass loading of ash in the atmosphere at the time of the over-pass. Using the calibrated emission rate we can also assess the total mass of ash remobilised over the entire event. The calibration applied here is uniquely related to the event studied and the source areas defined, but our approach can be used to consider how the emission rate of resuspension is varying with time since the ash was deposited (see response to Reviewer 1).

We have improved the explanation of our approach in Section 4, modifying paragraphs 1 and 2 to read:

*'Here we determine the scaling coefficient (K) for the emission rate (F, Eqn 1) in the resuspension scheme in NAME. As we have data from only one OPC instrument we are unable to perform a robust calibration with surface $PM_{10}$ data. Instead we perform a calibration using the total column mass loadings of the remobilised ash cloud retrieved from VIIRS.'*

*'Figure 11 shows the frequency of binned total column mass loadings from the satellite retrievals and the NAME modelled mass loadings where K is set to unity (1 g $s^{-1}$). The mode of the VIIRS mass loadings varies with time during the event, from $10^{-1} – 10^{0}$ g $m^{-2}$ to $10^{0} – 10^{1}$ g $m^{-2}$, this variation includes the uncertainty associated with the retrieval. The modelled total column mass loadings have a mode at $10^{-4} – 10^{-3}$ g $m^{-2}$. Considering the difference in the mode of the VIIRS retrieved mass loadings and the model output at each retrieval time suggests we need to apply a scaling of between K = $1 \times 10^{3}$ - $1 \times 10^{4}$ to the emission rate in the resuspension scheme in NAME to match the observed mass loadings in the atmosphere.'*

We have also made the following modifications to make the discussion of our aims and approach more consistent and clearer throughout:

Modified Line 14 in the abstract, to state that: we calibrate the *emission rate* in the resuspension scheme, rather than the *source strength*.

In the Introduction we now clearly state our aim and approach:

Page 3, Line 3: *'The newly calibrated scheme can be used to provide more accurate quantitative forecasts of future events and assess how resuspension rates are varying over time.'*

Page 3, Line 21: *'In Section 4 we calibrate the emission rate in the resuspension scheme in NAME with the satellite retrieved total column mass loadings and quantify the total mass of ash resuspended during 16 - 17 September 2013.'*

In Section 3.1 on the modelling approach we now state that $K$ is set to unity when the emission rate is un-calibrated on Page 6 Line 4:

*'Without calibration K is set to 1 g s$^{-1}$'*

We have changed *'source strength'* to *'emission rate'* on Page 6 Line 5.

On Page 6 Line 11 we now state that:

*'Figure 3 shows the time series of calculated air concentrations from OPC count data at Mariubakki for the period 9 September to 2 October 2013. Modelled air concentrations using the un-calibrated emission rate (K=1 g s$^{-1}$) are compared.'*

On Page 6 Line 32 we state that:

*'The edge of the ash cloud is identified as 1-hour averaged mass loadings > 1 X 10$^{-7}$ (g m$^{-2}$), with this threshold taken as a pragmatic plotting choice as the emission rate is un-calibrated.'*

Finally in the Conclusions section we have modified *'source strength'* to *'emission rate'* on Page 15 Line 16.

The section on water vapour, whilst technically correct, is completely undermined by the final section of 3.1.1 where the discussion grinds to the halt as it is explained that the water vapour correction was not applied. Recast this section to explain what was done (in more detail) rather than a more complex explanation of something that wasn't. Why does the BTD signal in Figure 8 get stronger at lower mass (for constant particle size). That is opposite to what every paper I've ever read on the subject would suggest. There may be other reasons for positive BTD. The authors should probably approach them, and rule them out (especially coating of the ash and/or mixing with ice). I accept that this is unlikely but there are precedents in Iceland, though not from re-suspended ash.

Response: As stated in the paper, water vapour may affect the measured BTD signal. We thus find it important to discuss the water vapour effect for the case studied here as it is significantly different for other cases in the literature. We show that to include a water vapour correction is not trivial when ash is located at the same altitudes as the water vapour. Thus we choose to not include a water vapour correction before ash pixel identification. However, in the look-up table calculations used in the retrieval, area averaged ECMWF water vapour profiles, as described in Section 2.3 (Supplementary Figs. S1–S4), were used. To more precisely describe this, the last paragraph of section 3.2 has been rewritten and now reads:

*'The 16-17 September 2013 resuspended ash cloud had a top height of about 1.0 km (Fig. 7). As is evident from Fig. 8 and the discussion above, any water vapour correction for an ash cloud at this altitude is not straightforward. Thus, no water vapour correction was applied before ash pixel identification. Rather, a customized ash detection scheme was applied, see next section. For the ash mass loading retrieval the absorption of water vapour was included in the look-up-table calculations using area averaged ECMWF water vapour profiles, see Section 2.3 and Supplementary Figs. S1-S4.'*

In Fig. 8 the BTD signal does not get stronger at lower mass. For zero ash (black line) the BTD is constant with altitude. Introducing ash changes the BTD at all altitudes. This change is the ash signal. We are not aware of papers showing this change for ash clouds at low altitudes, there are however numerous papers showing this for ash clouds at higher altitudes (see for example Wen and Rose, 1994; Prata and Prata, 2012). Thus let us qualitatively compare our results at say 8 km with their results. Increasing the ash mass loading from 0 to 0.01 g m$^{-2}$, decreases the BTD from about 0.5 to -9

K. Further increasing the mass loading increases the BTD until the signal in the two channels saturate (BTD is about 1.5 K). This bowl shaped behaviour is similar to the behaviour shown Fig. 2 of Wen and Rose (1994) and Fig. 2 of Prata and Prata (2012) for ash clouds at a fixed altitude.

The end of the first paragraph of section 3.2 has been rewritten to clarify this:

*'For an ash cloud at 8 km the $BTD_V$ decreases from about 0.5 to -9 K when the ash mass loading increases from 0 to 0.01 g m$^{-2}$. Further increasing the mass loading increases the $BTD_V$ until the signal in the two channels saturate ($BTD_V$ about 1.5 K). This bowl shaped behaviour is qualitatively similar to the behaviour shown in Fig. 2 of Wen and Rose (1994) and Fig. 2 of Prata and Prata (2012) for ash clouds at higher altitudes. Figure 8a further shows that for ash cloud top heights above 2.0 km, $BTD_V$ is negative for mass loadings less than 0.02 g m$^{-2}$. Contrary, $BTD_V > 0.0$ when the top of the ash cloud is between 0.5-2.0 km and mass loadings are $\geq 0.02$ g m$^{-2}$. As the 16-17 September 2013 resuspended ash cloud top is between 1-2 km a positive $BTD_V$ signal is therefore to be expected for volcanic ash, as seen in Supplementary Fig. S6.'*

We have added the following text to the end of Section 3.2 to discuss other reasons for a positive BTD:

*'It is noted that the presence of ice may give a positive BTD (see for example Rose et al., 1995). However, due to the ambient temperatures and the origin of the resuspended ash we rule out the presence of ice for the case studied here.'*

Leadbetter et al., 2012 proposed a range of 0.4 – 0.5 for U*t.  What difference would using 0.5 make (i.e. how sensitive to the final outcome is that choice)?

In Leadbetter et al. (2012) they show that using a threshold friction velocity of 0.4 m s$^{-1}$ is most appropriate for modelling the resuspension of PM$_{10}$, as when using a threshold of 0.5 m s$^{-1}$ resuspension events were missed and truncated, when compared to observed PM$_{10}$ count data. They also note that this agrees well with a threshold of 0.42 m s$^{-1}$ identified from wind tunnel experiments (Sigurjonsson et al., 1999). Further, Folch et al. (2014) have subsequently also shown that a threshold friction velocity of 0.4 m s$^{-1}$ was most appropriate when modelling the resuspension of fallout deposits from the June 2011 Cordon Caulle eruption in Central Patagonia during October 2011. We have no new evidence to suggest that we should not be using 0.4 m s$^{-1}$ and a full sensitivity test was beyond the scope of this work. We now clearly justify our choice of threshold friction velocity in Section 3.1, Paragraph 1:

*'Leadbetter et al. (2012) found that using a threshold friction velocity of 0.4 m s$^{-1}$ was most appropriate for modelling the resuspension of ash from deposits following the 2010 eruption of Eyjafjallajökull in Iceland. They also note that this agrees well with a threshold of 0.42 m s$^{-1}$ identified from wind tunnel experiments (Sigurjonsson et al., 1999). Folch et al. (2014) also found that a threshold friction velocity of 0.4 m s$^{-1}$ was most appropriate when modelling the resuspension of fallout deposits from the June 2011 Cordon Caulle eruption in Central Patagonia during October 2011.'*

What effect does limiting the NAME 1-10 microns have? Given much larger (and more mass bearing) particles have been observed (and discussed later in the paper) this is something the authors will need to further explore.

Response: We have extended our discussion in Section 5 to consider how our modelled mass loadings may vary if a larger particle size range were resuspended on Page 14 Line 5:

*'Liu et al. (2014) measured the PSD of resuspended ash deposited in Reykjavik during the 6 –7 March 2013 following a significant remobilisation event of the Eyjafjallajökull 2010 and Grímsvötn 2011 deposits. Most of the mass was contained within the 32 – 63 µm size fraction and < 10 % of the total mass was on particles with diameter < 10 µm. Here we have considered particles with diameter ≤ 10 µm only, to be consistent with the particle size range the satellite retrievals are most sensitive to. No observations of the PSD of the remobilised ash cloud were made during the 16-17 September 2013. Taking the PSD from Liu et al. (2014) suggests that our calculated remobilised mass of 0.2 Tg for this event may represent a fraction of the total mass actually resuspended.'*

In summary there are some quite unfathomable things in the paper. I would encourage the authors to work through these and provide explanations. It could be I've simply misunderstood but even that would imply a lack of clarity in the paper.

Editorial (very minor):

P1 L19 should be ':' not ';'

Fixed

P2 L10 'random walk' might need further explanation / reference

Response: The following references have now been added:

Maryon, R.H., Ryall, D.B. and Malcolm, A.L. The NAME 4 Dispersion Model: Science Documentation, Met O Turbulence and Diffusion Note No. 262, 1999.

Thomson, D.J. and Wilson, J.D., 'History of Lagrangian stochastic models for turbulent dispersion', In: Lagrangian modelling of the atmosphere, Geophysical Monograph Series 200, American Geophysical Union, Washington, pp. 19-36, 2013

P4 L1-5. Reasons for mass loadings are presented but do not mention reduced availability. This is then discussed later in the paper, foreshadow that discussion, briefly here (it seems to be to be a perfectly reasonable explanation, as, of course, does the location of the OPC).

Response: We now comment in Section 2.2 that only one OPC data-set is available to us for this event:

*'The lower mass loadings recorded during this event perhaps reflect the availability of data from only one OPC, which was not positioned under the main axis of the resuspended ash cloud, but instead was located at the edge of the plume....'*

P4 L19 Maybe quote the calculate dBTD cost from water vapour here (for the purists).

Response:  Quote included in the text.

P7 L1 This is clumsy. Do you mean your doubled the concentration of the lowest layer to preserve constant mass?

This sentence has been clarified. It now reads:

*'For the ash cloud with a maximum altitude of 0.5 km the ash concentration was doubled to preserve constant mass.'*

P11 L3 cite 'Mackie, S., Millington, S. and Watson, I.M., 2014.  How assumed composition affects the interpretation of satellite observations of volcanic ash. Meteorological Applications, 21(1),  pp.20-29.'?

Fixed

P12 L16 Chronologise reference list

Fixed

Figures look good in colour but are unusable in black and white (no change required unless the paper won't be published in colour)

We intend to publish in colour.

Additional Changes:

We have removed the following from Section 3.1 Line 6 as this was repeating the Introduction:

*'Once released into the model atmosphere particles are advected using the 3-dimensional NAE model winds and dispersed using random-walk techniques which account for turbulent structures in the atmosphere.'*

And moved the following sentence to the Introduction where the rest of the discussion on NAME is:

*'Particles are removed from the atmosphere by both dry and wet deposition processes (Webster and Thomson, 2011, 2014).'*

Finally, we have corrected the numbering of the sub-sections in Section 3, such that they are now 3.1, 3.2, 3.3 and 3.4. Previously they read 3.1, 3.1.1, 3.1.2 and 3.2.